# Detection of Candidate Genes Associated with Fecundity through Genome-Wide Selection Signatures of Katahdin Ewes

**DOI:** 10.3390/ani13020272

**Published:** 2023-01-13

**Authors:** Reyna Sánchez-Ramos, Mitzilin Zuleica Trujano-Chavez, Jaime Gallegos-Sánchez, Carlos Miguel Becerril-Pérez, Said Cadena-Villegas, César Cortez-Romero

**Affiliations:** 1Recursos Genéticos y Productividad-Ganadería, Colegio de Postgraduados, Campus Montecillo, Carretera Federal México-Texcoco Km. 36.5, Texcoco 56264, Mexico; 2Producción Animal, Universidad Autónoma Chapingo, Carretera Federal México-Texcoco Km. 38.5, Texcoco 56230, Mexico; 3Agroecosistemas Tropicales, Colegio de Postgraduados, Campus Veracruz, Carretera Xalapa-Veracruz Km. 88.5, Manlio Favio Altamirano, Veracruz 91690, Mexico; 4Producción Agroalimentaria en Trópico, Colegio de Postgraduados, Campus Tabasco, Periférico Carlos A. Molina, Ranchería Rio Seco y Montaña, Heroica Cárdenas 86500, Mexico; 5Innovación en Manejo de Recursos Naturales, Colegio de Postgraduados, Campus San Luis Potosí, Agustín de Iturbide No. 73, Salinas de Hidalgo, San Luis Potosí 78622, Mexico

**Keywords:** F_ST_, ROH, candidate genes, fertility, ovine

## Abstract

**Simple Summary:**

It is difficult to improve the reproductive performance of sheep (fecundity, fertility, and prolificacy) through the simple selection of animals that are outstanding due to phenotype. The genetic transmission of the reproductive performance to progeny is of low effectiveness for these types of traits, which is why traditional selection has not been successful. Presently, there is scarce information about the reproduction of Katahdin sheep and apparently no information about genes associated with reproductive traits. In this study, a genetic scan was conducted to detect genes associated with fecundity in Katahdin ewes. It was found that genes expressed in reproductive tissues tend to be reported frequently as candidates; that is, they are genes that directly influence the fecundity of ewes: *CNOT11*, *GLUD1*, *GRID1*, *MAPK8,* and *CCL28*. Other genes have indirect influence, for example, those that affect hormonal processes such as lipid synthesis: *ADIRF*, *CCL28,* and *HMGCS1*. New genes associated with the fecundity of Katahdin ewes are reported in this study.

**Abstract:**

One of the strategies to genetically improve reproductive traits, despite their low inheritability, has been the identification of candidate genes. Therefore, the objective of this study was to detect candidate genes associated with fecundity through the fixation index (F_ST_) and runs of homozygosity (ROH) of selection signatures in Katahdin ewes. Productive and reproductive records from three years were used and the genotypes (OvineSNP50K) of 48 Katahdin ewes. Two groups of ewes were identified to carry out the genetic comparison: with high fecundity (1.3 ± 0.03) and with low fecundity (1.1 ± 0.06). This study shows for the first time evidence of the influence of the *CNOT11*, *GLUD1*, *GRID1*, *MAPK8,* and *CCL28* genes in the fecundity of Katahdin ewes; in addition, new candidate genes were detected for fecundity that were not reported previously in ewes but that were detected for other species: *ANK2* (sow), *ARHGAP22* (cow and buffalo cow), *GHITM* (cow), *HERC6* (cow), *DPF2* (cow), and *TRNAC-GCA* (buffalo cow, bull). These new candidate genes in ewes seem to have a high expression in reproduction. Therefore, future studies are needed focused on describing the physiological basis of changes in the reproductive behavior influenced by these genes.

## 1. Introduction

In Mexico, sheep (*Ovis aries*) are abundant in arid and temperate zones, where their meat is an important food source for families of scarce resources and consumers of traditional and festive dishes. However, more information is needed about the species to improve the reproductive characteristics of economic interest, such as fertility [1].

Fertility, fecundity, and prolificacy are the three key reproductive traits to attain sustainable sheep production [2,3]. Fecundity is the ability of animals to produce progeny and it is important in production systems, given that it impacts directly other traits such as prolificacy, number, and kilograms of weaned lambs and the total kilograms of meat produced [2,3,4]. Although in Mexico sheep of the Katahdin breed are not as abundant as those of the Pelibuey breed, the first stand out for their tolerance to heat and their high growth speed and weight gain [5,6]. In addition, it is a rustic breed with a high tolerance to parasites in comparison to wool sheep [6], and it has a prolificacy of 1.3 [7] with double and triple births [5].

Reproductive traits in sheep are of low inheritability, which is why selection by phenotype results in scarce annual genetic gain [8]. The identification of candidate genes is one of the strategies to improve these traits, and these genes are known as fecundity genes or *Fec* genes [9]. Three important genes have been identified that affect fecundity in ewes: *BMPR1B* gene or *FecB* (bone morphogenetic protein receptor, type 1B), *BMP15* gene or *FecX* (bone morphogenetic protein 15), and *GDF9* gene or *FecG* (growth differentiation factor 9) [9,10]. The *GDF9* gene is expressed in the luteum tissue and in all the development stages of the ovarian follicle, and it directly influences the prolificacy of ewes; in addition, it has been shown that a total absence of its expression leads to infertility of the ewes [11].

In addition to the major genes, other genes associated with fecundity have been detected in different breeds of sheep, such as: *PDGERL*, *FSHR*, *LEPR*, *KLF5,* and *PDGFRL* (selection signature detection) [12]. Some other genes associated with prolificacy are: *FLT1*, *CCL2* (classic GWAS, or Genome-Wide Association Study) [8], *ODZ1*, *ODZ3*, *LTBP3*, *DSCAM* [13], *ESR1*, *GHR*, *ETS1*, *MMP15*, *FLI1*, and *SPP1* (classic GWAS) [14].

Because of this, it is necessary to explore more genetic mechanisms for fecundity in ewes through the identification of candidate genes. Given the genetic differences between breeds, the application of a GWAS is pertinent to find genes associated with fecundity [15]. There are different GWAS methodologies to find candidate markers: the classical GWAS or marker by marker [16]; the use of machine learning [17,18]; the retrospective association analysis [18]; and the most novel molecular methodology, the use of selection signatures [2,19].

Artificial (or natural) genetic selection has left a footprint in the genomes of animals; these footprints are known as selection signatures, which provide information about the domestication and evolution processes that resulted in the animal species and breeds currently known. There are selection signatures based on the construction of haplotypes that measure the similarity between DNA segments of two populations, such as haplotype homozygosity extended between populations (XP-EHH) and the hapFLK statistics. Other types of signatures based on the individual measure of marker diversity are fixation index, Tajima’s D, and runs of homozygosity (ROH) [20].

ROH are selection signatures widely used for candidate gene detection in small ruminants [20]. ROH are contiguous lengths of homozygous genotypes surrounding a favorable mutation [12,20] and are evidence of inbreeding. This kind of selection signature is used to detect haplotypes with high homozygosity throughout the population. In sheep it has helped to find candidate genes [12].

The objective of this study was to detect candidate genes associated with fecundity through the fixation index (F_ST_) and run of homozygosity (ROH) of selection signatures in Katahdin ewes.

## 2. Materials and Methods

### 2.1. Ethical Declarations

Based on the regulations for the use and care of animals destined to research at Colegio de Postgraduados [21], 0.5 mL of blood was collected per ewe by puncture of the jugular vein with sterile syringe, under the criteria of the NOM-062-ZOO-1999 for technical specifications for the production, care, and use of laboratory animals [22]. These blood samples were used for genotyping of animals.

### 2.2. Phenotypes

Records from three years (2019–2021) of the number of offspring birthed per ewe by mounting were used as a measure of the fecundity of 48 Katahdin ewes from the Agricultural and Livestock Production Unit “Quinta San Francisco”, found in Hidalgo, Mexico, with coordinates 19°54′56″ N, 98°40′12″ W and altitude of 2500 m [23], with average age and number of births of 4.48 ± 1.28 years and 3.2 ± 1.31, respectively. In addition, the year of registry, age in years, body condition (1–5 scale), hours to estrus, and number of births per ewe were observed.

For the size of the sample used, the potency of the statistical test was determined using the *pwr* package [24] of R [25]. The entry values of the function in R were the size of the sample (48), the significance (0.5) and the method (Xi^2^).

To perform the statistical correction of the phenotypes, a logistic regression was used with the Gaussian model, with the use of the *glm* function of the *stats* package of R version 4.2.0 [25]. The year of the record, age in years, body condition, hours until estrus, and number of births were used as independent variables, and were tested via the Wald Xi^2^ test, and from them only the intercept of the model and body condition were statistically significant. The response variable was fecundity, measured as the number of offspring per mounting per year per ewe (0, 1 or 2 lambs). According to the results of the Wald Xi^2^ test, only the intercept of the model and body condition were incorporated in the adjustment of the final model (*p* < 0.05):$$y=\beta_{0}+\beta_{1}x_{1}$$
where y is the annual fecundity, $\beta_{0}$ is the intercept of the model, $\beta_{1}$ is the estimator associated with body condition, and x is the body condition. The adjustment of the model obtained an accuracy of prediction of 75.4%. The phenotypes adjusted with this model were used to carry out the rest of the analyses.

To verify the presence of two subpopulations in the population of Katahdin ewes (high and low fecundity), a hierarchical cluster analysis was conducted. The hierarchical grouping analysis was carried out using the Euclidian distances matrix of the phenotypes adjusted by the model, through the *stats* package [25], through the *hclust* and *cutree* functions, to classify the ewes into high and low fecundity. Finally, a circular dendrogram was constructed from the previous results using the circlize *dendogram* function of the *dendextend* R package [26], to observe 23 ewes classified as “high fecundity” and 25 as “low fecundity”.

A t-test was conducted to determine if there were differences (*p* < 0.05) between the means of fecundity in the subpopulations created; the t.*test* function of the *stats* package [25] for groups with non-homogeneous variance was used.

### 2.3. Genotypes

Blood samples from 48 Katahdin ewes were genotyped with the Illumina OvineSNP 50K chip (NEOGEN, Lincoln, NE, U.S.A., https://www.neogen.com/, accessed on 23 March 2022). For the quality control of genotypes (51,867 markers), the following single nucleotide or punctual (SNP) polymorphisms were eliminated: with a level of frequency of allele lower than 0.05 (3838), those that were not in Hardy–Weinberg equilibrium (516; *p* > 0.000001), and those with a call percentage lower than 90% (261). The final number of markers after quality control was 47,084 SNP.

### 2.4. Population Structure

To observe the correspondence of the phenotypic and genotypic divergences of the population, a principal components analysis (PCA) was conducted through the *adegenet* R package [27]. The entry variables for dimension reduction were the 47,084 SNP markers and the fecundity (high or low) of the ewes was used as a supplementary variable.

### 2.5. Selection Signatures

To find the different proportions of the genome with contiguous homozygous genotypes (ROH), the PLINK 1.07 software was used [28] to create the *ped* and *map* files. Later, the *detectRUNS* package of R [29] was useful to find the proportions of the homozygous genome. Because these proportions also differed in length and type between the groups of ewes of high and low prolificacy, the function used was *slidingRuns* for the method of sliding windows with 50k of opening. The values for the *slidingRuns* function parameters were: windowSize = 15, threshold = 0.05, minSNP = 20, maxOppWindow = 1, and maxMissWindow = 1.

With the plot_*manhattanRuns* function of the same package, a Manhattan graph was built that shows the SNP that are most frequently detected within a ROH by subpopulations. Haplotypes formed through ROH above 75% of frequency and made up by three or more markers were considered as candidates.

On the other hand, F_ST_ fixation indexes were obtained for both subpopulations with the use of the *pegas* [30] and *adegenet* packages [27]. To identify the SNP related to fecundity, the difference between the fixation values per SNP between the subpopulations was calculated. Markers with the highest 20% were considered as candidates.

### 2.6. Detection of Candidate Genes

To find the genes associated with the candidate SNP markers, the position and the chromosome were considered. Genomic regions with a range of ±50 K pb around the candidate SNP were considered as candidate regions with reference to the genome (https://www.ncbi.nlm.nih.gov/assembly/GCF_000298735.2, retrieved on 15 June 2022); the candidate genes were detected as those found in these regions.

## 3. Results

### 3.1. Population Structure

For the sample size of 48 animals and *p* = 0.05, the power of the test was 0.7. The graphic representation of the PCA is shown on the left side on Figure 1. The points in the PCA represent the ewes under study, located in function of their genotypic divergence, while the ellipses represent the groups phenotypically formed according to the fecundity of the ewes. Two differentiated genetic groups were observed from the animal population, which in addition agree with the phenotypical classification given by the supplementary variable, fecundity. The mean for high fecundity of ewes was 1.3 ± 0.03 lambs per mounting, while for low fecundity it was 1.1 ± 0.06 lambs per mounting (*p* < 0.05). The circular dendrogram is shown on the right side of Figure 1, which was built from Euclidian phenotypical distances, which illustrate the significant difference (*p* = 3.3 × 10^−13^) between the subpopulations of sheep of high (blue) and low (red) fecundity.

### 3.2. Candidate Genes

Candidate markers were found for the ROH and F_ST_, 79 and 14, respectively. The details of the results for the ROH are shown in Table 1; only candidate haplotypes containing more than three SNP were considered. Table 1 also shows the start and end of the ROH segments. The total length of the genome, considering only 25 chromosomes, was 2.784 billion bp. The total ROH were 569 for the high fecundity group and 552 for the low fecundity group.

According to the information from the Ovis_aries_v4.0 genome, a total of five candidate genes associated with low fecundity and 10 with high fecundity were obtained, through the ROH method (Figure 2). On the other hand, with the F_ST_ method, seven other candidate genes associated with high fecundity were found (Table 2), different from those detected by the ROH method. 

Table 2 shows the candidate genes found in this study associated with fecundity, which includes previous reports of associations with other reproductive characteristics. Meanwhile, Figure 2 shows the Manhattan graph for high and low fecundity with the five and three principal candidate genes, respectively, identified by the ROH method (Table 2), with effects on the fecundity of Katahdin ewes and in reproductive traits.

In this study, 17 genes associated with high fecundity and five with low fecundity were observed. Once these genes were selected through selection signatures (F_ST_ and ROH), they were filtered according to functions previously reported as being associated with reproduction in sheep, cattle, deer, horses, and pigs (Table 2).

## 4. Discussion

### 4.1. Population Structure

The power of the test of 0.7 is acceptable to obtain significant real differences. Including this test allowed validating the results, since the in-mass genotyping of animals is not available [54]. This impacts the few populations of the Katahdin breed that are present in Mexico, given that they limit the application of new genomic methodologies for their genetic improvement.

In genome-whole association studies and studies of genetic diversity, the use of multivariate analysis is common to characterize the genetic and phenotypic structure of the populations. The most used methods are PCA [9,55,56] and metric multidimensional scaling [57,58]. As in other studies with sheep [15,49,59], in this study, the PCA made it possible to graphically observe the genetic and phenotypic differences of Katahdin ewes. Most of the candidate genes detection studies are based on two clearly divergent groups of animals: lines [57], breeds [55], and groups formed by a phenotype of a quantitative trait (such as the number of horns) [59]. In this study, we used individuals from the same population and created two groups based on their fecundity using multivariate analysis as in other studies [58]. As it was expected, the population structure in our study was different from other studies in which the divergence among groups is evident in the PCA or in the multidimensional scaling. Nonetheless, we found that the divergence used in this study was enough to detect candidate genes associated with fecundity.

### 4.2. Candidate genes

The gene *CNOT11* or *CCR4-NOT* transcription complex subunit 11 was reported as the candidate gene for pregnancy rate [31]. The gene expression in ewes takes place primarily in the uterus, the placenta membranes, and the embryo; in addition, together with the genes *SDHA*, *PPIA*, *RPS9,* and *RPL19*, it is one of the most stable marker genes in reproductive and fetal tissues [53]. In this study, with the use of the F_ST_ method, it was found that the *CNOT11* gene is a candidate for fecundity in Katahdin ewes, with a possible influence in the success of embryo formation and the culmination of pregnancy [53].

The *ANK2* (ankyrin 2) gene is known for its effect on productive characteristics. In bovines, it affects the meat and carcass quality [60], and in sheep it has been proven that it has an impact on the characteristics associated with growth, such as weight at weaning and finalization weight, as well as quality of the wool [61]. In sheep, no previous studies are known about the effect of the *ANK2* gene on reproduction, but they are in pig reproduction [34].

Contrary to what was found in this study, there are reports in sows of the negative impact of the *ANK2* gene on the structure of the granulose cells, which consequently have a negative effect on fecundity [34]. In this study, it was found that the *ANK2* gene is associated with high fecundity in ewes. The profile of expression of the *ANK2* gene on tissues of the sow has been scarcely studied, although it seems to have a different profile from that of the ewe. In the sow, the expression of the *ANK2* gene on the ovary is higher than in the ewe [62,63]. On the other hand, the expression of the *ANK2* gene in ewes is abundant in tissues of the neuro-endocrine system, such as the pituitary, the hypothalamus, the cerebellum, and the cerebrum [62]. Probably, the reproductive effects of *ANK2* differ between ewes and sows, due to the difference in tissues where they have their highest expressions. In ewes, the effect of *ANK2* could be due to changes in the secretion and segregation of hormones, while in sows it could be due to a local effect related to the functioning of the ovary and sexual organs [62,63].

No previous reports that associate the *HERC6* gene (HECT and RLD domain containing E3 ubiquitin protein ligase family member 6) with reproductive traits in ewes were found; however, there are reports of associations with growth, composition of the carcass, body size, weight, height, and milk production [64]. It was found that the *HERC6* gene, together with the *PARP12*, *RNF213* and *ZNFX1* genes, is involved in the early gestation of bovines (day 18) [49]. The main role of these genes, regulated by the *IFNT*, is the immune response to establish and maintain the uterine receptivity to implantation [49].

In our study, the *HERC6* gene, detected through ROH, was a candidate gene for low fecundity, which suggests the confirmation of its effect in the early gestation in sheep as well as in bovines. That this gene is strongly fixed in ewes with low fecundity could be indicative of its negative effect on the species, since it seems that the profile of gene expression in both species is the same [49,62], with a significant presence during the first 20 days of gestation in uterus and cervix.

The *CCL25* and *CCL28* genes (C-C motif chemokine ligands 25 and 28) were found with high expression during the early fetal development in sheep [50]. In particular, the *CCL28* gene is associated with the recruitment of immune cells in the spleen and the nasal mucosa of the fetus, which leads to the development of a functional immunological system pre-birth. In addition, there is evidence of the influence of the gene during the transfer of antibodies during lactation [50]. The *CCL28* gene also influences the gastric immunity of animals in any stage of life [50]. In our study, *CCL28* turned out to be a candidate gene for low fecundity, which signals a negative relationship between the innate level of immunity of the offspring and the number of lambs born per ewe per mounting.

This study confirms the *DPF2* (double PHD fingers 2), *ARHGAP22* (Rho GTPase activating protein 22), and *GHITM* (growth hormone inducible transmembrane protein) genes as candidates for fecundity in Katahdin ewes. In bovines, the *DPF2* gene was reported as a candidate for fertility and reproductive traits [65]; in addition, the expression of the gene in sheep is high in sexual tissues, such as testicles, uterus, and ovaries [62]. On the other hand, the *ARHGAP22* gene was reported in many species as a candidate for feminine reproductive traits [39]; the expression of the gene happens primarily in the placental membranes and in the ovary [62], which suggests an effect on the persistence of gestation in sheep. The *GHITM* gene has been associated with the persistence of metritis and reproductive traits in Holstein cows; it was found that it plays a defining role in the immune response of the organism in the presence of a bacterial infection (metritis), particularly in the cycle, the metabolism, and the cell communication [40]. In addition, there is evidence of high expression of the *GHITM* gene in the luteum body, ovaries, and uterus [62].

In this study, candidate genes were found for high fecundity with effects on reproductive traits that have been reported few times in the literature. The *GLUD1* or glutamate dehydrogenase 1 gene was associated with the metabolism of carbon that happens during development and follicle maturation in sheep [41], although there are no reports of their levels of expression in organs related with reproduction; this gene can have a relevant effect on the fecundity of Katahdin ewes. Another gene without reports of expression in sheep is the *TRNAC-GCA* or transfer RNA cysteine (anticodon GCA) gene; in bovines, this gene affects sperm quality [47]. In this study, it was found that the *TRNAC-GCA* gene is associated with high fecundity in Katahdin ewes, which confirms the importance of the gene in reproductive traits, at least in bovines and sheep.

The mitogen-activated protein kinase 8 (*MAPK8*) gene is important in ovine reproduction. In this study of association, its effect on the high fecundity of Katahdin ewes is confirmed. The *MAPK8* gene has a high expression in organs associated with reproduction, such as ovaries, pituitary, embryo tissues, and testicles [62]. It is one of the most studied genes in sheep, since it participates in the physiological processes of the ovary, particularly in the cells of the granulose, where it is part of follicle development, and consequently affects traits associated with fertility and with fecundity and prolificacy [44,66].

One of the genes associated most with reproductive traits in sheep is the glutamate ionotropic receptor delta type subunit 1 or *GRID1* gene [42]. In this study, it was associated with high fecundity of Katahdin ewes, while in other studies it was associated with size of the litter [42] and fertility [12]. The expression of the *GRID1* gene in sheep seems to happen in organs of the neuroendocrine system, particularly in the hypothalamus and the cerebrum, and there are reports of its expression in testicles and ovaries in lower numbers [62].

In this study, we considered other genes as candidates, which have not been previously related with reproductive traits, but whose function in the organism could indirectly influence the physiological processes that lead to reproductive phenomena. For example, F_ST_ made it possible to determine that the *ATG10* [autophagy related 10] and *RPS23* [ribosomal protein S23] genes are candidates for high fecundity in ewes; however, the main action of the genes happens in the immune system, which could impact indirectly the fecundity of the ewes in this study [32,33,41].

On the other hand, candidate genes for the metabolism of fat and energy can also be indirectly related to reproduction (as was found in this study), given the lipid composition of some reproductive hormones such as testosterone, progesterone, and prostaglandin. Thus, the genes in this situation are: *CAMK2D* [35] (calcium/calmodulin dependent protein kinase II delta), *STK32B* [36] (serine/threonine kinase 32B), *HMGCS1* [52] (3-hydroxy-3-methylglutaryl-CoA synthase 1), *ADIRF* [38] (adipogenesis regulatory factor), *LRIT1* [43], and *UGT8* [37] (UDP glycosyltransferase 8).

## 5. Conclusions

This study is the first to find candidate genes for fecundity in Katahdin ewes in Mexico. These candidate genes were detected through the selection signatures F_ST_ and ROH, which confirmed the influence of the five genes *CNOT11*, *GLUD1*, *GRID1*, *MAPK8,* and *CCL28* on the characteristics of fertility in sheep; in addition, six other new candidate genes were detected that had not been previously reported for this species: *ANK2* (sows), *ARGHGAP22* (cows), *GHITM* (cows), *HERC6* (cows), *DPF2,* and *TRNAC-GCA* (cows). The 17 new candidate genes for fecundity in sheep, found in this study, seem to have a high expression in reproductive organs. However, studies are necessary to elucidate the physiological basis of the changes in reproductive behavior influenced by these genes.

## Figures and Tables

**Figure 1 animals-13-00272-f001:**
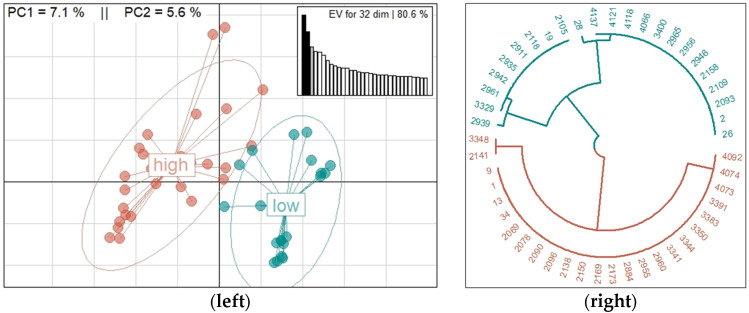
Graphic representation of the principal components analysis (PCA; (**left**)) and dendrogram built from Euclidian distances (**right**). Both representations show the subpopulations of Katahdin ewes in agreement with their level of fecundity (high or low). For the PCA, the Eigenvalues (EV) of 32 dimensions are shown whose sum of variances is equal to 80.6%, in addition to the variance explained by the first (PC1) and second (PC2) dimension.

**Figure 2 animals-13-00272-f002:**
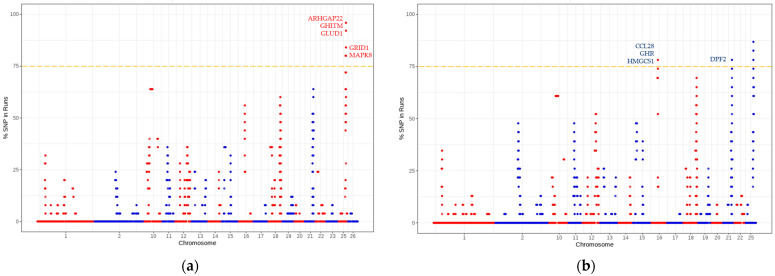
Genomic distribution of positive selection signatures for high (**a**) and low (**b**) fecundity using the frequency of SNP in the different ROH haplotypes in Katahdin ewes. The yellow dotted line indicates the minimum frequency required to consider ROH/SNP as candidates (75%).

**Table 1 animals-13-00272-t001:** Information of detected runs of homozygosity (ROH) in the genome of Katahdin sheep.

Fecundity Group	Start SNP	End SNP	Chromosome	Number of SNP	Position (bp)	
					From	To
Low	oar3_OAR6_36225598	oar3_OAR6_36295143	6	6	36225598	36295143
Low	Chr16:31054761	Chr16:31867174	16	11	31054761	31867174
Low	OAR16_31491759.1	OAR16_31817206.1	16	3	31491759	31817206
Low	oar3_OAR16_31358348	oar3_OAR16_31643375	16	3	31358348	31643375
Low	Chr21:42744589	Chr21:42752272	21	3	42744589	42752272
High	Chr25:36456595	Chr25:37312419	25	20	36456595	37312419
High	oar3_OAR25_36631465	oar3_OAR25_37179818	25	7	36631465	37179818
Low	Chr25:35575408	Chr25:35687973	25	5	35575408	35687973
Low	Chr25:35770158	Chr25:35919902	25	8	35770158	35919907
Low	Chr25:36456595	Chr25:36800459	25	11	36456595	36800459
Low	OAR25_35590985.1	OAR25_36580394.1	25	3	35590985	36580394
Low	oar3_OAR25_35598041	oar3_OAR25_35873731	25	5	35598041	35873731

**Table 2 animals-13-00272-t002:** Candidate genes putatively selected by two statistical methods of selection signature detection affecting fecundity in Katahdin ewes and reproduction traits reported in other studies.

Method	Gene	Chromosome	Position	Function Summary
High fecundity				
F_ST_	*CNOT11*	3	100058810	Pregnancy rate [31].
	*ATG10*	5	79156925	Autophagy and innate immune defense [32,33].
	*RPS23*	5	79446021	Protein synthesis and immunity [9].
	*ANK2*	6	12314476	Structure of ovarian granulosa cells [34].
	*CAMK2D*	6	11870774	Fat metabolism and meat quality traits [35].
	*STK32B*	6	103213333	Fat deposition and marbling [36].
	*UGT8*	6	11004274	Development and function of the nervous and endocrine systems [37].
ROH	*ADIRF*	25	41056054	Adipose deposition tissue [38].
	*ARHGAP22*	25	42255625	Fertility traits, lipid metabolism, insulin resistance, and inflammation response [39].
	*GHITM*	25	38626280	Metritis and fertility traits [40].
	*GLUD1*	25	41085197	Follicular development and maturation [41].
	*GRID1*	25	39879600	Litter size [42].
	*LRIT1*	25	38706733	Cellular energy [43].
	*MAPK8*	25	42210625	Follicular development [44].
	*MMRN2*	25	41018603	Cell differentiation or organogenesis [45].
	*TRNAC-GCA*	25	39208014	Plasma urea concentration and sperm quality [46,47].
	*VSTM4*	25	42748968	Cumulus cells and somatic cell nuclear transfer [48].
Low fecundity				
ROH	*HERC6*	6	36181502	Pregnancy rate [49].
	*CCL28*	16	31346400	Lymphocyte colonization of fetal tissues [50].
	*GHR*	16	31832298	Growth performance, carcass traits, milk traits, and cell differentiation [51].
	*HMGCS1*	16	31409821	Cholesterol biosynthesis [52].
	*DPF2*	21	42741418	Fertility [53].

## Data Availability

Data are available from first author Reyna Sánchez-Ramos, <rey_1014@hotmail.com> upon request.

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
