# Peer review of "Detection of Candidate Genes Associated with Fecundity through Genome-Wide Selection Signatures of Katahdin Ewes"

_animals, 2023, doi:10.3390/ani13020272_

Round 1
Reviewer 1 Report
The present manuscript provides important findings hoverer, the methodology and results are not defined properly. Generally, the manuscript has focused on the genes and not the way they were identified.
Introduction
Lines 69-71, You describe the association of the genes with fecundity but not the method that was used to detect it. It should be included.
Include in the text the way in which those genes will be used in practical breeding (i.e., via MAS).
Enrich the introduction about the ROH: what is it, in which animals has this approach been used in the literature. Have any important findings (i.e., genes) associated with economical traits been found using this method/technique?
A short paragraph describing the aims of the study is helpful for the reader. Please, determine your tasks clearly.
Materials and Methods/Phenotypes
By “adjustment of the phenotypes” (line 109) do you mean “corrected phenotypes”. Did you use the statistically significant fixed effect to correct the phenotypes?
In the model (line 109-115), the response/dependent variable was the average of offspring per mounting per year per ewe, which is a continuous variable. Why did you use a logistic regression?
(Line 109-115) The year of the record, age in years, body condition, hours until estrus, and number of births are used as independent variables and are tested via Wald Xi2 test, and from them only the intercept of the model and body condition were statistically significant; If I get it right, please enrich the paragraph with all the steps. Otherwise, correct me and rewrite it.
The t-test (Line 123) is used as validation of the clustering which took place, not as a grouping method. Please, correct it to avoid any misunderstanding.
This module is very important for the present study because the creation of the subpopulations was based on that. I think that you should enrich it with the steps which are missing and describe it more extensively.
Materials and Methods/Selective signatures
For the ROH, basic information is missing. Please describe with more details the steps, for example, the widowsize threshold, minSNP, maxMissWindow. Also, are the ROH grouped per length; provide more information about the length category average. Which percentage of the chromosomes has been “covered” by the ROH; Number of ROH per chromosome. Did you also find the common ROH between the groups; All those findings are missing from the text as well as from the results.
The description of the Fst in Lines 157-159 is confusing. Based on my knowledge, the Fst is a technique to evaluate the degree of difference between two groups per SNPs, which agrees with the lines 157-158.5. But I do not understand the second part, can you describe more clearly?
Materials and Methods/ Detection of candidate genes
Which software did you use to find the genomic regions around the candidate markers?
Results
For table 1, please include the start and end position of the ROH along with the length of it. How many SNPs are in the ROH in each genomic region and the start and end position of the gene. Also, the frequency of the ROH.
Figure 2. Explain what the yellow line in the label represents. Is this figure representing all the groups or only the high fecundity? If it is, please also include the low fecundity group. Furthermore, the diagram for the Fst which compares the two groups must be added.
Also, provide more information about the ROH. As well as information about the ROH islands.
Discussion
A comparison between other breeds of sheep and the Katahdin by using the ROH, (i.e., higher average ROH, higher cover of chromosomes etc) or other genes related to fertility have been found in other breeds of sheep by using the ROH / GWAS etc.
Is there any other study which categorizes the sheep in two groups based on fertility? If there are/is, are they also finding genetic diversity between their groups, or are the same fixed effects statistically significant?
Author Response
December 17th, 2022.
Subject: Correction of manuscript ID animals-2085364-R1
Ms. Ann An
Assistant Editor/MDPI
ANIMALS
E-Mail: ann.an@mdpi.com
Dear Ann,
I am sending (attached) the manuscript entitled "Detection of candidate genes associated to fecundity through genome-wide selective signatures of Katahdin ewes."
We greatly appreciate the valuable comments and suggestions made by the editor and reviewers to this manuscript.
The reviewer(s) and editor´s comments have been covered and all changes highlighted in green were included throughout the document.
Comments from the Assistant Editor
(I) Please check that all references are relevant to the contents of the manuscript.
(II) Any revisions to the manuscript should be marked up using the “Track Changes” function if you are using MS Word/LaTeX, such that any changes can be easily viewed by the editors and reviewers.
(III) Please provide a cover letter to explain, point by point, the details of the revisions to the manuscript and your responses to the referees’ comments.
(IV) If you found it impossible to address certain comments in the review reports, please include an explanation in your appeal.
(V) The revised version will be sent to the editors and reviewers.
Comments from reviewer 1
The present manuscript provides important findings hoverer, the methodology and results are not defined properly. Generally, the manuscript has focused on the genes and not the way they were identified.
Introduction
Lines 69-71, You describe the association of the genes with fecundity but not the method that was used to detect it. It should be included.
Response: Thank you for your recommendation. The information has been added in the 4th paragraph of the introduction section, highlighted in green, lines 71-73.
Include in the text the way in which those genes will be used in practical breeding (i.e., via MAS).
Response: We appreciate your recommendation; nonetheless, we consider that information not necessary and even can be confusing. In this study we identified candidate genes associated with a complex trait such as fecundity. At the moment, we have not the ambition to use the found candidate genes to select animals (MAS). This is the first study finding candidate genes in the Katahdin sheep; however, more studies are necessary to determine the approach in practical breeding.
Enrich the introduction about the ROH: what is it, in which animals has this approach been used in the literature. Have any important findings (i.e., genes) associated with economical traits been found using this method/technique?
Response: The information has been added in the 7th paragraph of the introduction section, highlighted in green, lines 90-93.
A short paragraph describing the aims of the study is helpful for the reader. Please, determine your tasks clearly.
Response: The aim was improved in the 8th paragraph of the introduction section, highlighted in green, lines 94-96.
Materials and Methods/Phenotypes
By “adjustment of the phenotypes” (line 109) do you mean “corrected phenotypes”. Did you use the statistically significant fixed effect to correct the phenotypes?
Response: Sorry for this evident mistake. It has been modified (Line 116) in the third paragraph of the phenotypes section (Mat and Met) and highlighted in green. We corrected the phenotypes by the body condition instead (as is mentioned in the manuscript), which it was the only significant effect.
In the model (line 109-115), the response/dependent variable was the average of offspring per mounting per year per ewe, which is a continuous variable. Why did you use a logistic regression?
Response: The word “number” has been added in the third paragraph of the phenotypes section (Mat and Met) and highlighted in green. Logistic regression has been used due to the nature of the data. It is correct that we obtained an average per sheep to build the fecundity groups; nevertheless, we had several repetitions by same sheep through the years, so the data ranged between the integers values 0, 1, and 2; which is a non-continuous variable.
(Line 109-115) The year of the record, age in years, body condition, hours until estrus, and number of births are used as independent variables and are tested via Wald Xi2 test, and from them only the intercept of the model and body condition were statistically significant; If I get it right, please enrich the paragraph with all the steps. Otherwise, correct me and rewrite it.
Response: Thanks for your help to improve the clarity of our manuscript (Lines 118-123). This observation has been added in the third paragraph of the phenotypes section (Mat and Met).
The t-test (Line 123) is used as validation of the clustering which took place, not as a grouping method. Please, correct it to avoid any misunderstanding.
Response: It has been done (Line 134).
This module is very important for the present study because the creation of the subpopulations was based on that. I think that you should enrich it with the steps which are missing and describe it more extensively.
Response: There are not many details to give. We think the procedure of t test is really simple and no more information is needed.
Materials and Methods/Selective signatures
For the ROH, basic information is missing. Please describe with more details the steps, for example, the widowsize threshold, minSNP, maxMissWindow. Also, are the ROH grouped per length; provide more information about the length category average. Which percentage of the chromosomes has been “covered” by the ROH; Number of ROH per chromosome. Did you also find the common ROH between the groups; All those findings are missing from the text as well as from the results.
Response: Thanks. The information for the ROH has been added in the first paragraph of the selective signatures section (Mat and Met). Also, we do not include the information of ROH in common between the groups, because our objective was always to find the differences between high and low fecundity sheep (Lines 163-166), then we consider it not necessary.
The description of the Fst in Lines 157-159 is confusing. Based on my knowledge, the Fst is a technique to evaluate the degree of difference between two groups per SNPs, which agrees with the lines 157-158.5. But I do not understand the second part, can you describe more clearly?
Response: You are right. That information has been corrected and added in the third paragraph of the selective signatures section (Mat and Met), lines 174-175.
Materials and Methods/ Detection of candidate genes
Which software did you use to find the genomic regions around the candidate markers?
Response: We downloaded the information of genes positions from NCBI, and with R software, we determined the candidate genes by comparing them with the candidate markers positions.
Results
For table 1, please include the start and end position of the ROH along with the length of it. How many SNPs are in the ROH in each genomic region and the start and end position of the gene. Also, the frequency of the ROH.
Response: Thank you for your valuable recommendation. The Table 1 has been added (Lines 203-208) to detail the information about detected ROH.
Figure 2. Explain what the yellow line in the label represents. Is this figure representing all the groups or only the high fecundity? If it is, please also include the low fecundity group. Furthermore, the diagram for the Fst which compares the two groups must be added.
Response: The ROH Manhattan plot (with the description for the yellow line) has been included (Figure 2). Nonetheless, we consider not necessary the inclusion of the FST diagram, because we only focused on the high fecundity candidate genes. Also, only one low fecundity candidate gene for FST has been discovered, a gene that has no antecedents of association with the trait in the study.
Also, provide more information about the ROH. As well as information about the ROH islands.
Response: The information has been included in the Table 1.
Discussion
A comparison between other breeds of sheep and the Katahdin by using the ROH, (i.e., higher average ROH, higher cover of chromosomes etc) or other genes related to fertility have been found in other breeds of sheep by using the ROH / GWAS etc.
Response: The study was focused on the discovery of candidate genes. Thus, in the discussion, we tried to describe what has been found about the genes more than the ROH segments, because a region in the genome is less comparable among breeds than the genes that have an effect on the trait. We discussed some important points among breeds, when it was possible, due to the limited information reported.
Is there any other study which categorizes the sheep in two groups based on fertility? If there are/is, are they also finding genetic diversity between their groups, or are the same fixed effects statistically significant?
Response: We only found a research with a methodology similar to this study. But they did it using different breeds, and in this study we just used Katahdin breed. Study: Nosrati et al. (2018). Whole genome sequence analysis to detect signatures of positive selection for high fecundity in sheep. DOI: https://doi.org/10.1111/rda.13368
Answers to reviewer 2
Katahdin sheep breed description is missing. Information in the second paragraph of Introduction is very general not providing any view on why this breed has been selected for research, nor advance to the cosmopolitan ones to be of interest of readers. What is the ecnomic importance of the breed and population size/ structure?
Response: Such as it was mentioned at the beginning of the introduction section, in Mexico, the Katahdin breed is important for a group of people. Actually, the breed is not well-known across the world but has local level importance, so there are no relevant studies characterizing such breed, which is the reason we cannot give a lot of information about the breed.
2.2 Missing clear presentation on the structure of phenotypic data used for analyses.
Response: It has been done in the third paragraph of the phenotypes section (Mat and Met) and highlighted in green. We changed some phrases to make it more clear (Lines 116-134).
Why adjusted phenotypes instead of originaly recorded were used? Where is the explanation of the sole significance of body condition in the model?
Response: It is well known that before performing a genetic study over a population, a phenotypic adjustment is needed due to the environmental effects which can disguise the pure genetic effect. Then, we tested different independent variables to see which of them was pertinent to use in the model for the phenotypes adjustment. Some changes have been done (Lines 116-125).
How many SNPs were eliminated by your current setting of MAF in the QC?
Response: It has been done in the paragraph of the genotypes section (Mat and Met) and highlighted in green.
What is your explanation on level of MAF applied?
Response: It is not necessary to establish the reasons, because is the threshold used globally.
Results could be limited by sample size and "statistical adjustment" of the phenotypic data.
Response: The results definitely are limited due to such situations you mentioned and we are aware of that. For that reason, a power test has been done (Lines 184-187) and the adjustment of phenotypes must be limited. Thus, we prefer to have fewer candidate genes if those genes are more precisely detected.
Terms:
inheritance/ inheritability - how they differ with well recognized inheritance/ heritability
Selection signatures vs. selective signatures ... again the first one is well established.
I don´t fully support idea of innovation of terminology when not needed.
Response: Thank you. We have made some changes in the text, which were highlighted in green.
Sincerely yours,
On behalf of all the co-authors
Ph.D. (Dr.) César Cortez-Romero
Researcher & Professor
Tel. +52 496 963 04 48 ext. 4000
+52 595 952 02 00 ext. 75044
E-mails: ccortezro@hotmail.com; ccortez@colpos.mx

Reviewer 2 Report
Katahdin sheep breed description is missing. Information in the second paragraph of Introduction is very general not providind any view on why this breed has been selected for research, nor advance to the cosmopolitan ones to be of interest of readers. What is the ecnomic importance of the breed and population size/ structure?
2.2 Missing clear presentation on the structure of phenotypic data used for analyses.
Why adjusted phenotypes instead of originaly recorded were used? Where is the explanation of the sole significance of body condition in the model?
How many SNPs were eliminated by your curent setting of MAF in the QC?
What is your explanation on level of MAF applied?
Results could be limited by sample size and "statistical adjustment" of the phenotypic data.
Terms:
inheritance/ inheritability - how they differ with well recognized inheritance/ heritability
Selection signatures vs. selective signatures ... again the first one is well established.
I don´t fully support idea of innovation of terminology when not needed.
Author Response

(The authors gave the same response as above.)

Round 2
Reviewer 1 Report
The present manuscript provides important findings and the methodology is better described than in the first version although the results are not illustrated thoroughly.

Author Response
Please see the atachment

Reviewer 2 Report
N/A
